# The Garden and the Orchestra: Generative Metaphors for Conceptualizing the Complexities of Well-Being

**DOI:** 10.3390/ijerph192114544

**Published:** 2022-11-05

**Authors:** Tim Lomas, Tyler J. VanderWeele

**Affiliations:** Harvard T.H. Chan School of Public Health, Human Flourishing Program, Harvard University, Boston, MA 02115, USA

**Keywords:** flourishing, well-being, health, metaphor, conceptualization

## Abstract

Our understanding of well-being, and related concepts such as health and flourishing, is shaped by the metaphors through which we think about such ideas. Current dominant metaphors—including a pyramid, ladder, and continuum—all have various issues. As such, this paper offers two other metaphors which can better do justice to the nuanced complexities of these notions, namely, a garden and an orchestra. Through these metaphors, this paper articulates a comprehensive framework for conceptualizing and appreciating the nature of well-being (and associated concepts), which it is hoped will generate further insights and research into these valued and sought-after phenomena.

## 1. A Metaphorical Approach to Conceptualizing Well-Being

In recent decades, we have seen a surge of academic interest in well-being, and related topics such as health and flourishing. However, despite this burgeoning attention—or possibly because of it—there is currently little agreement about what these terms mean. Indeed, among the many reviews of definitions and conceptualizations of these concepts [1,2,3,4,5,6], these invariably emphasise how elusive and contested these are, and relatedly how difficult to find consensus. From a UK policy context, for example, a 2008 report from the Department for Children, Schools and Families lamented “there is significant ambiguity around the definition, usage and function of the word ‘well-being,’” with the discourse “at present particularly unstable” [2]. Fifteen years on, the situation has not much improved. Reviewing trends in conceptualizations of well-being and related concepts in a chapter for the 2022 World Happiness Report, Barrington-Leigh suggests that while the use of these terms has risen sharply, these are “typically poorly defined” [6], p. 1. Despite these concepts having been embedded into policy and even law, their actual meaning often remains rather vague or at least under-theorized. As such, this paper offers a way of conceptualizing these concepts which can provide clarity, and furthermore, does justice to their nuanced complexity.

In particular, we harness the power of metaphor to help us better understand and appreciate these ideas. This is because metaphor is not only a useful way of conceptualizing complex ideas, but is central to thought itself. This point was brought to widespread attention by Lakoff and Johnson [7], whose ground-breaking 1980 book Metaphors We Live By has over 80,000 citations as of October 2022. Their work—and the extensive scholarship inspired by it—highlights the intimate way thought and language are shaped by our embodied experience in the world. As Lakoff [8] writes, our conceptual systems “grow out of bodily experience… [being] directly grounded in perception, body movement, and experience of a physical and social character” (p. xiv). In their theory, in exploring the world as infants, our embodied experience generates three main types of schema: (a) spatial orientations (e.g., up-down); (b) ontological concepts (e.g., container); and (c) structured experiences (e.g., moving). These then provide the basis for a rich, complex system of metaphorical concepts. Such concepts also take three main forms, each drawing primarily on one of the main schemas. Orientational metaphors allow people to think abstractly about phenomena using spatial dynamics (e.g., “rising levels of happiness”). Ontological metaphors confer entity or substance status onto phenomena, for example, describing one’s mind as a container (“full of thoughts”). Finally, structural metaphors allow abstract activities (e.g., understanding) to be configured as more concrete ones (e.g., “I see what you mean”). Given these insights, we might consider what the main metaphors are for conceptualizing well-being at present, and how well do these work (i.e., how fruitful, accurate, and interesting are they).

Perhaps the most well-known is Maslow’s hierarchical “pyramid” of needs [9]. This holds that people have six universal needs—physiological, safety, love and belonging, esteem, self-actualisation, and self-transcendence—with well-being depending, at least partly, on one’s environment meeting these. His innovation was to conceive of these hierarchically: lower levels are more fundamental, taking priority over higher ones, and if not adequately satisfied, then higher ones diminish in importance. Everyone wants respect, for instance, but if one is starving, *that* dominates one’s concerns. However, as generative as his theory has been, there are issues with its central metaphor. Interestingly though, there is no record of Maslow himself developing the iconic pyramid configuration with which his work is associated; Bridgman et al. suggest it may have first been formulated in the 1960s by Charles McDermid in the context of adapting Maslow’s theory as a tool for management consultants [10]. Indeed, Maslow disavowed the central implication of the pyramid—especially when configured as a sequence of steps—that one might ascend “up” the hierarchy by fulfilling, thus leaving behind, its foundational levels, writing that most people are “partially satisfied in all their basic needs and partially unsatisfied in all their basic needs at the same time” [11], p. 388. This point has been corroborated by subsequent research, which shows, for example, that higher-level resources can mitigate the relative lack of lower ones. Meaning and purpose, for instance, can be achieved even amidst material deprivation; indeed, efforts to address such deficiencies may *provide* people with purpose, levels of which are often higher in countries with greater economic and material difficulties [12]. Given such issues, Kaufman [13] instead suggested using a sailboat metaphor, which does seem a better way of representing Maslow’s theory—lower levels represent the hull, constituting the existential security that keeps us afloat, while the higher levels are the sail, which provides the dynamic movement that enables exploration, personal development, and ultimately, self-actualization.

After Maslow, perhaps the next most prominent and influential metaphor is a ladder. This stems principally from Cantril’s Self-Anchoring Striving Scale [14], which invites people to envisage a ten-rung ladder—whose base and top, respectively, represent the worst and best possible life imaginable—and indicate where they stand. The item is still widely used, including in the annual Gallup World Poll, one of the most comprehensive worldwide surveys, whose results are the basis for the World Happiness Report [15]. Despite its prominence though, the metaphor is also problematic in various ways. For a start, it is inherently unidimensional, so struggles to capture the multidimensional nature of well-being, as we explore below. That said, the ladder is intended as a global assessment (all elements together) of one’s life generally, and a person may well make all kinds of multidimensional calculations in determining the rung on which they presently stand. Even so, the metaphor does not allow representation of the complexity of well-being, where a person can be doing well in some respects but less well in others at any given time. A further issue is that the ladder represents what we might call a “maximalist” view of well-being, whereby the higher one climbs, the better. While that may seem a natural way of appraising well-being in Western cultures, it is not necessarily universally shared. There are suggestions, for example, that Eastern cultures may not aspire towards this maximalist vision, with people potentially wary of being situated *too* high on the ladder, for various complex reasons. This includes, for example, a possible “fear of happiness,” involving a belief that happiness may have negative consequences, such as increasing the likelihood one will be unhappier in the future [16,17]. In that context, it has been suggested that Eastern cultures prefer a “balanced” view of well-being, where rather than being situated at the extremes—even in a positive way, at the top of the ladder—it is better in many ways to inhabit some optimal point between the poles [18]. We shall return to such ideas below.

Besides the pyramid and the ladder, perhaps the next most well-known metaphor is a continuum or spectrum of well-being. This presents well-being as arrayed upon a linear configuration, spanning a negative and positive pole—sometimes taken as representing illness and health, respectively—with a neutral and nominal “zero” at the mid-point. Among the earliest invocations of this idea was, again, by Maslow [11]. This occurs in the context of critiquing Freud, and psychology more broadly, for mainly focusing on the negative territory of the spectrum. Indeed, Freud acknowledged this himself, observing that the goal of psychotherapy was generally limited to turning “hysterical misery into ordinary unhappiness” [19], p. 308—i.e., helping people reach the relative neutrality of zero. While that is an important goal, Maslow sought to emphasise that just because a person is ostensibly free of suffering, this does not mean they are actively thriving, living their best life, and fulfilling their potential. He thus advocated exploring the positive terrain in establishing what came to be known as humanistic psychology, saying “It is as if Freud supplied us the sick half of psychology and we must now fill it out with the healthy half” (p. 5). This spectrum metaphor was subsequently embraced by positive psychology (PP), since despite the efforts of Maslow—and others such as Jahoda [20], who propounded the notion of “positive mental health”—the adherents of this new field felt that psychology was still mainly focused on the negative territory. Thus, this new field sought to bring renewed focus to the positive realm of the spectrum as, for example, influentially articulated by Keyes in his paper The Mental Health Continuum: From Languishing to Flourishing in Life [21]. 

However, as generative as this metaphor has been, it suffers from similar issues as the ladder, such as unidimensionality. Indeed, Keyes himself recognised this limitation, arguing that the negative and positive realms of mental illness and health are actually disconnected—physiologically, functionally, and experientially—and that people can experience aspects of illness and health at the same time. As such, Keyes [22] embraced the idea of a “dual continua” model, with separate continua for illness and health, which visually could be placed orthogonally to create a bivariate state space, with a horizontal axis of illness, and a vertical axis of health. However, while this is an improvement on a unidimensional spectrum, it is still limited, since it is increasingly recognized that illness and health are not singular either, but themselves contain multiple continua—such as different forms of mental illness (e.g., depression, anxiety, etc.) or mental health (e.g., happiness, meaning, etc.)—with any given person concurrently doing poorly on some and better on others. Yet a further challenge to the continuum metaphor, or any metaphor, is that certain aspects of well-being might be conceived of as being genuinely bipolar (such as the spectrum from loneliness to social connectedness), whereas others might be conceived of as unipolar (such as meaning, wherein a sense of meaninglessness really is simply constituted by the absence of meaning). Complicating the picture further, meaning, social connectedness, and loneliness are themselves each arguably multi-dimensional [23].

Thus, there are issues with all the current dominant metaphors used to conceptualize well-being and associated ideas. Thus, here we explore the potential for new metaphors that might better reflect their dynamic complexities and nuances, namely, a garden and an orchestra. Although we are not the first to note their creative potential in relation to well-being [24,25], to our knowledge, they have not been harnessed and elaborated upon in the kind of depth we seek to do here. Each is interesting and generative in different ways, and could serve equally well in doing greater justice to well-being and its associated multidimensional concepts. We introduce these metaphors over several sections, in the course of which we, (1) offer a definition of well-being (and related notions such as health and flourishing), (2) draw attention to their nuanced dynamics, and (3) situate flourishing as the more all-encompassing term, including health and well-being within. 

## 2. Defining Well-Being

Before delving into the metaphors themselves, it will help to first outline the way we define well-being, as well as related concepts such as health and flourishing. Like many scholars, we regard these as multidimensional. Before specifying these dimensions though, we can first define well-being in generic terms. In doing so, we must also conceptualize the closely related notion of health, since these two terms are often used in conjunction with one another—one commonly encounters the phrase “health and well-being”—in ways that suggests that together they capture different and complementary aspects of a common phenomenon. Defining these is tricky, because in contemporary discourse, one can find, (1) the terms used synonymously, (2) health presented as a subset of a broader notion of well-being, and (3) the terms overlapping, often in ways that are not well specified. In that context, our prerogative in defining these terms is twofold: (1) to identify distinct conceptual roles for each (i.e., rather than using them interchangeably), while also (2) honouring the spirit of the diverse ways in which these are used in various definitions and in the broader discourse. 

Let us begin by noting that both health and well-being have two main roles, denoting: (a) a conceptual space pertaining to how well a person is faring in life, and (b) the desirable aspect/area of that space. An analogy for the way they possess this dual function is the concept of height. This is likewise an overarching concept, referring to one’s vertical orientation in space, encompassing specific positional terms such as short and tall. However, when describing something as “high,” this nevertheless positions it towards the top range of values within the range denoted by “height.” The same double meaning also applies to health and well-being. In that respect, we can offer subtly different definitions for both (a) and (b), using the same key words, that apply to both health and well-being. Thus, we define (a) as the quality of one’s personal state, and (b) as the relative attainment of a personal state of quality. Significantly, our priority is with (b), so whenever we use either health or well-being, it is this second definition we have in mind. However, one may encounter other definitions that use these more in the sense of (a), and our dual definition framework here can also accommodate these usages.

Let us briefly clarify the key terms in these definitions. First, we use “personal” to differentiate health and well-being from flourishing, which is personal *and* systemic: as explored below, while health and well-being concern a specific individual, flourishing extends both to human people and to their myriad contexts—both human (e.g., communities) and non-human (e.g., the natural environment)—in which they are situated. We deploy “state” to signify a condition or mode of being that is not permanent, but can vary widely in duration (from a fleeting emotion of seconds to a durable way of being that could even last for years). Finally, we harness the idea of “quality,” as deployed in notions of quality of life, a common framing in work on well-being [26]. This idea has further been meaningfully explored by Pirsig [27], for whom it described a fundamental and irreducible sense of a phenomenon being deemed good or valuable in some way [28].

At this point, we are still using health and well-being synonymously. However, we would ideally find distinct conceptual roles for them. In that respect, the most common way we can identify in which these appear to be differentiated in the literature—even if scholars themselves did not explicitly invoke the following binary—is in terms of the distinction between ontological subjectivity (i.e., phenomenal qualia) and objectivity (i.e., material or non-qualitative processes and states). That is, whenever health and well-being are not being used synonymously, health often tends to denote objective aspects of the person (i.e., how well their body and brain are functioning), while conversely, well-being frequently seems to emphasize subjective aspects of the person (i.e., how they *feel*, whether physically or mentally). Here, we shall embrace this distinction. Thus, both (a) and (b) definitions above—where (a) denotes a conceptual space pertaining to how well a person is faring in life, and (b) the desirable aspect/area of that space—can be adapted to emphasize either objectivity for health or subjectivity for well-being. As such, health refers to (a) the quality of one’s personal objective state, and (b) the relative attainment of a personal objective state of quality, while well-being refers to (a) the quality of one’s personal subjective state, and (b) the relative attainment of a personal subjective state of quality. Again, when we use either term, our priority is with (b), so it is these second definitions we have in mind. In that respect, one can also use ill-health and ill-being for the undesirable aspects/areas of these conceptual spaces, contrasting health with ill-health and well-being with ill-being. Thus, adapting the (b) definitions to describe lack rather than attainment, ill-health could be construed as the relative lack of a personal objective state of quality, while ill-being could be construed as the relative lack of a personal subjective state of quality.

A crucial caveat to note at this point is that not all definitions in the literature align health and well-being, respectively, with objectivity and subjectivity in this way. Instead, in some definitions, these terms cover both objective and subjective aspects of the person, which is of course also fine (even if it is not *our* prerogative) This ambiguity applies to some of the most prominent formulations, including most notably the WHO’s (1948) definition of health as “a state of complete physical, mental and social well-being, and not merely the absence of disease and infirmity” [29]. As such, when approaching such existing definitions, we would not want readers to impose our subjective–objective distinction on them. Instead, one must use judgement in recognizing that, in such definitions, health and well-being are being used more interchangeably, covering both objectivity and subjectivity. 

So far, we have been discussing and conceptualizing health and well-being in generic terms. However, further nuance can be brought to our understanding of this broad terrain by applying and adapting these formulations to different dimensions of existence. There are many ways of conceptualising this complex ontological terrain, but one common framing is to acknowledge physical, mental, and social dimensions of existence, as per the WHO’s definition of health above. In addition, some scholars suggest that we ought to recognize spiritual well-being [30,31]. Together, we refer to these four dimensions as the “WHO+” ontological framework [32]. We can, therefore, adapt the generic definitions of health and well-being to apply to these dimensions specifically. Thus, for example, one can describe physical health as (a) pertaining to the quality of one’s personal objective physical state, and (b) its attainment as a personal objective physical state of quality, while physical illness could be defined as a personal objective physical state lacking quality. Similarly, one could describe physical well-being as (a) pertaining to the quality of one’s personal subjective physical state, and (b) its attainment as a personal subjective physical state of quality, while physical ill-being could be defined as a personal subjective physical state lacking quality. One could similarly adapt these for the other dimensions by inserting the relevant adjective (e.g., mental, social, or spiritual) instead of physical.

Moreover, if health and well-being are attained across *all* dimensions, this would constitute complete or overall health and well-being. Thus, for health, we would define its remit (a) as the quality of one’s personal objective state across the physical, mental, social, and spiritual dimensions of existence, and its attainment (b) as a personal objective state of quality across the physical, mental, social, and spiritual dimensions of existence. Then, for well-being, we would similarly define its remit (a) as the quality of one’s personal subjective state across the physical, mental, social, and spiritual dimensions of existence, and its attainment (b) as a personal subjective state of quality across the physical, mental, social, and spiritual dimensions of existence. Finally, most optimal of all would be complete health *and* well-being across all dimensions, whose remit (a) is construed as the quality of one’s personal objective and subjective state across the physical, mental, social, and spiritual dimensions of existence, and whose attainment (b) is a personal objective and subjective state of quality across the physical, mental, social, and spiritual dimensions of existence. 

With these definitions in mind, we can bring these ideas to life through our two metaphors, which we will build up over the course of the paper. From this point on, please note that although we are interested in both health (as an objective state) and well-being (as a subjective state), to avoid cluttering the text, we shall just refer to well-being. Thus, even though well-being technically refers to a subjective state in our definitions, when encountering this term in the text below, please interpret it as also referring implicitly to health (i.e., to the counterpart objective state). Equally though, if one preferred to emphasise objective rather than subjective states, one could adapt the text below to refer instead to health rather than well-being; in that case, we would suggest that although health technically refers to an objective state in our definitions, when encountering this in the text, one might interpret it as also referring implicitly to well-being (i.e., to the counterpart subjective state).

## 3. New Well-Being Metaphors

Having defined our key terms, we now turn to the two new metaphors we suggest can help us better understand well-being (and related concepts). The first is a garden. Let us start by imagining each human being as a tree (or alternatively, a plant, bush, or flower). In terms of overall well-being, clearly, we are interested in developing this organism into its full potential, so it becomes the best possible version of itself. We can also recognize that the tree has numerous aspects—each symbolising one of the dimensions of human well-being—all of which play their role in this process. One might imagine the tree’s roots as constituting physical well-being, these being the foundation from which it draws its strength and anchors its existence. Its trunk and branches represent mental well-being, these being the overall form and shape of the tree, just as mental functioning determines the very way in which people experience and understand their existence. Its flowers and fruit symbolise social well-being, these being the essence of its interaction with other trees—and various life forms—in the form of cross-pollination and reproduction facilitated by creatures such as bees. Finally, its leaves signify spiritual well-being, in that they draw energy and nourishment from a higher power (i.e., the sun) through the process of photosynthesis. Moreover, as per the dimensions of human well-being, these various elements are both somewhat independent, yet also intrinsically interrelated. Just as a human might excel physically but struggle mentally, for instance, one could imagine a tree that had strong roots but a misshapen structure (e.g., due to being damaged by a storm). Yet, overall, with humans and trees, the different dimensions and elements tend to co-evolve and function together as a whole. Thus, for instance, a human with good physical and mental well-being is more likely to also enjoy good social well-being than someone struggling in those dimensions. Likewise, a tree with strong roots and structure is liable to produce better flowers and leaves than one lacking these solid foundations, while from the other direction, the leaves effectively receiving nourishment from the sun are likewise important for the roots and structure of the tree itself. 

In developing this metaphor, it can firstly reflect the idea that—in each ontological dimension—the territories of ill-being and well-being are both multifaceted, not a single continuum. This not only means that ill-being and well-being are separate domains—as reflected in Keyes’ dual continua model [22]—but in themselves can have various forms, with any given person doing better or worse on all these at a given time. With the mental domain, for instance, there are myriad forms of mental ill-being; indeed, the DSM-5 includes 157 different disorders [33]. Moreover, these are usually diagnosed on the basis of composite symptoms; the criteria for major depression, for instance, are that a person must experience five or more out of eight symptoms during a two-week period (with at least one symptom being depressed mood or loss of interest/pleasure). Similarly, in terms of positive mental well-being, there are numerous different forms. It is common to differentiate hedonic and eudaimonic forms of mental well-being, though even these are multifaceted; Ryff’s notion of “psychological well-being”, for example, which pertains to eudaimonia, has six dimensions [34]: self-acceptance; positive relations; autonomy; environmental mastery; purpose in life; and personal growth. Then, in addition to these are other potential forms, such as “engaged well-being” [35], which includes phenomena such as flow [36], as well as various aspects of character which is itself highly multidimensional [37,38]. All these different aspects of mental ill-being and well-being could be imagined as each constituting a separate branch of the tree. Aspects of mental ill-being or well-being that are unipolar might be pictured in terms of the presence and relative length of a specific branch, whereas aspects that are bipolar might be viewed in terms of a branch that is clearly part of the tree but in a state of relatively more or less decay or health. This metaphor could be as granular as one wished. Less granularly, for example, there could be one branch for each mental disorder and one for each main form of mental well-being. Being diagnosed with the disorder would mean that a particular branch was “lacking quality,” whereas not having the disorder would mean it did have quality; conversely, not having that particular aspect of mental well-being would also mean that branch lacked quality, while having it meant the branch had quality. More granularly, one could have different branches for all the myriad symptoms and aspects of these forms of ill-being and well-being. Then, overall, the quality of one’s mental state would be reflected in the overall quality of the branches collectively. 

We shall continue to develop the garden metaphor in the next section. First, though, we can also begin to introduce the orchestral metaphor. An orchestra of course involves numerous musicians arranged into different units based on the type of instrument played (e.g., woodwind, strings, brass, etc.), and our metaphor will eventually include this entire collective. However, let us start by imagining an individual human as a single instrument being played. As with the garden metaphor, in terms of overall well-being, we are concerned with this being played to its full potential, becoming the best possible version of itself. Similarly, we can also view this process as having numerous aspects, each symbolising one dimension of human well-being. Thus, physical well-being is symbolised by the quality of the instrument itself—how well designed, made, looked after, etc.—as the foundation for even being able to make sounds at all. Mental well-being is signified by the relative quality of the melody the instrument is playing; just as mental functioning fundamentally shapes people’s existence, the tune the instrument is actually playing is perhaps its most dominant and significant contribution. Then, social well-being is denoted by the quality of its coordination and harmony with other instruments, i.e., how well the instrument fits with others in the orchestra and contributes to its overall sound. Finally, spiritual well-being is reflected in how well the instrument is being played—the quality of the musician and their performance in the moment, i.e., whether a note is being played at precisely the right time in the right way with appropriate dynamics—reflecting the way the instrument receives its energy and impetus from a source outside itself (i.e., the musician). As with the garden metaphor, these elements are somewhat independent, yet also intrinsically interrelated. An instrument could be well-made and playing an attractive melody, for instance, yet the playing could still be poor and out of synch with other instruments. However, it is more common for the various aspects to rise or fall together (e.g., a good musician will often have a good instrument). Indeed, to really reach the heights of musical performance, all four aspects will be excelling together.

Moreover, as with the garden metaphor, this too can be elaborated upon to reflect the potential co-existence of different forms of ill-being and well-being within any given dimension. Let us again use the mental domain as an example, as symbolised by the melody played by the instrument. The various forms of mental ill-being and well-being could each be specific notes within the overarching melody. Ill-being would be signified by notes of poor quality for various reasons (e.g., discordant or out of tune) and well-being by notes of good quality (e.g., elevating the melody aesthetically, being perfectly pitched). This metaphor also allows the possibility of a given form of ill-being and well-being dominating a person’s experience; just as a person might be mired in major depression for some time, imagine trying to play a melody on an instrument in which a particular string is broken or an organ key is jammed, discolouring the whole movement. However, even then, other notes can still make an appearance, just as moments of mental well-being can break through the depression. Moreover, over time and with the benefit of hindsight and an expanded context, it may be possible for experiences of mental difficulty to even be appraised positively in certain ways, just as a challenging and discordant passage of music can sometimes be seen as having a certain aesthetic value within the context of an entire piece. We can delve further into such ideas in the next section, which focuses on the nuances of well-being.

## 4. The Nuances of Well-Being

Central to the notion of well-being is its distinction from ill-being (and likewise, the difference between ill-health and health), as outlined above. However, the difference between these is not always self-evident or clear-cut, and there are numerous subtleties to consider. Perhaps the most pertinent issue is the significance of valence. This can be regarded as an “evaluative response” to one’s current situation, the “operations by which organisms discriminate threatening from nurturant environments” [39], p. 401. This is often conceptualised in terms of approach versus withdrawal. Positively valenced emotions are linked to neurophysiological and behavioural attraction towards a stimulus, with concomitant feelings of pleasure and reward. Conversely, negatively valenced emotions are associated with reaction against a stimulus, with associated dysphoric feelings. However, just as ill-being and well-being are not a single continuum, so too with valence. Rather than a single bipolar affective mechanism, Cacioppo and Berntson’s Evaluative Space Model [39] presents the affect system as a bivariate space, in which positive and negative valence are functionally independent, and indeed may sometimes be experienced simultaneously in the form of “ambivalent” or “mixed” emotions [40]. Moreover, the key question here is, what is the relationship between valence and ill-being/well-being? One sometimes encounters the assumption that negative valence is associated with ill-being and positive valence with well-being; in the context of the mental domain, this was initially the dominant perspective within PP, for example [41]. However, a wealth of emerging research has now challenged that assumption, a body of work referred to as “PP 2.0” [18] or “second wave” PP [42]. This literature indicates that some negatively valenced experiences can actually be valued as states of quality—such as sad aesthetic experiences, as considered below—hence, are construed as aspects of well-being. Conversely some positively valenced states might be deemed as lacking in quality (e.g., manic episodes), hence, aspects of ill-being. 

This brings up complex questions of whether and when certain states should be deemed forms of ill-being. Consider sadness as an example. We noted above that prolonged sadness (or “low mood”) is one symptom of depression. Given that a diagnosis of depression requires symptoms be present for two weeks, one might thus potentially situate extended sadness (i.e., longer than two weeks) as a form of ill-being. However, the psychiatric literature is riven with debates on the appropriateness of such judgements. In the case of grief for example, many critics suggest that some extended period of sadness is normal, and hence, not a disorder in any conventional sense of this word [43]. Similarly, even apart from grieving, Horwitz and Wakefield have argued against the medicalization of “regular” sadness, instead advocating regarding it as an inherent part of the human condition [44]. Going further, emergent literature suggests that negatively valenced emotions such as sadness can even be valued as integral to well-being [45]. For example, new work shows some people express preference for a “psychologically rich life,” which involves diverse and complex experiences and emotions, including negatively valenced ones [46]. Consider for instance that people often choose to engage with music and other media that evokes sadness [47]. Numerous explanations have been advanced, including emotional regulation and catharsis, aesthetic value and communicative intent, and appreciation of ambivalent/mixed emotions, sometimes referred to as “sweet sorrow” [48]. 

Going further, scholars such as Wong have argued for the value not only of negatively valenced emotions, but what could even be considered the more strongly qualified notion of suffering [18]. Drawing on traditions like Buddhism, Wong argues that suffering and strife are inherent to the human condition, and cannot be eluded. However, he draws solace and inspiration from Frankl [49], whose creation of logotherapy—healing through meaning—derived from his suffering in World War II. Frankl powerfully argued that meaning and redemption can be found even amidst the difficulties of life. Going further, such difficulties may even be *sources* of well-being. This insight rests on the recognition that it is people’s very experience of suffering that may make their happiness so precious and valued, a dark backdrop that allows the light to shine all the brighter. This insight was articulated poetically by Khalil Gibran, who wrote, “The deeper that sorrow carves into your being, the more joy you can contain” [50], p. 36. Similar insights can be found in the burgeoning work on the notion of post-traumatic growth, defined as “positive changes in the aftermath of crisis” [51]. Various aspects of the Christian tradition likewise teach that suffering can sometimes be a pathway to greater character, hope, and spiritual development [52]. There are also echoes in the literature on personal development, in which making progress may require a person to step outside their comfort zone, which may feel inherently challenging but ultimately may be integral to such progress. However, this realisation that suffering can potentially offer a paradoxical pathway to well-being does not necessarily come easily or quickly to people, but may require hard-won experience, hence, Wong describing it as a form of “mature happiness.” 

Moreover, going deeper into notions of valence and the phenomena associated with them, it can even become difficult to categorically assign certain experiences as positively or negatively valenced at all, since such appraisals are context-dependent. A vital context, for example, is time and the changing internal dynamics of the person. A certain stimulus may initially be positively valenced, yet without changing can, over time, become negatively valenced as the person themselves is altered by the experience. Consider someone who is hungry, and has a liking for sweet foods, craving an ice cream. Initially, having some would be deeply pleasurable on various dimensions (physically, mentally, etc.), experienced as refreshing and delicious. However, suppose they were forced to keep eating, way past satiation. That very same ice cream would soon be perceived as overly sweet and distasteful, and the experience unpleasant and negatively valenced; it has changed from a state of quality to one lacking quality. The general point is that not only may forms of well-being involve negatively valenced states, even the very distinction between positive and negative valence can often be nuanced and context-dependent.

A further strength of our metaphors is their ability to accommodate these nuances. With trees and other flora, for example, there is often a seasonal rhythm to their flowering, whereby for most species the overall lifetime well-being of the organism depends as much upon times of non-flowering (e.g., in winter) as flowering (e.g., in summer). Likewise, the development of a healthy overall plant, especially in the context of an overall healthy garden—as we consider below in the section on flourishing—may require seemingly paradoxical activities such as pruning, which afflict some ostensible short-term damage on the plant for the sake of its long-term prospects. Here, we return to the difference between “maximalist” versus “balanced” conceptualizations of well-being noted above. From the former perspective, if something is good, then more of it is necessarily better. However, there is emerging literature on the centrality to well-being of notions of balance and harmony [53]. Moreover, although these concepts are often associated with Eastern cultures, new research on these topics through the Gallup World Poll suggests they are valued and experienced in all cultures [54,55]. Similar ideas appear in Aristotle [37], in which virtue is often pictured as skilfully treading a fine line between two opposing vices. Such principles are integral to the garden metaphor; common sense tells us that, for all flora, while growth is normal and healthy, unrestrained growth is not healthy. This is certainly the case for the garden as a whole—as we consider below—where such lack of restraint constitutes a danger to other flora. It may also often be the case for the organism itself; a tree that has grown too tall for its environment, or a branch that has extended too far, may make the tree more unstable and liable to fall. Inherent in the notion of a thriving tree—and a flourishing garden overall—is it developing to just the right size, striking a balance between being under- and over-developed.

Similar nuances are accommodated by the orchestra metaphor. To begin with, there is an obvious association between positive and negative valence and major and minor keys, respectively. Indeed, this is not merely symbolic; research has shown the literal connection between these phenomena; minor keys, for example, are widely perceived as inherently sad [56], although this linkage might be moderated to an extent by cultural conditioning [57]. However, as noted above, and as any music lover will know, people frequently listen to, appreciate, and even—possibly paradoxically—deeply enjoy sad music [47]. More broadly, consider how the best music, while always high quality, nevertheless involves myriad intricate dynamics—not only minor and major, but also quiet and loud, slow and fast, etc. Likewise, well-being involves a dynamic, enriching mixture of experiences. Just as a piece involving nothing but major chords at full volume might get boring, so too might an emotional life characterised only by maximal hedonic pleasure. Indeed, as noted above with the ice-cream example, valence can have some strange psychodynamics whereby if pleasure persists without changing, it can actually become unpleasant. In any case, although one might assume people would rarely choose to experience negatively valenced emotions, studies into engagement with sad music—and other emotional art forms, such as challenging films—suggest that people actually do often seek experiences that may elicit such feelings. Likewise, so does the aforementioned research on people’s preferences for a psychologically rich life [46]. Moreover, even if people do not seek such states per se, if it serves their deeper and longer-term well-being, work on post-traumatic growth suggests that, from a broader perspective, they may come to appreciate their quality. 

## 5. The Broader Context of Flourishing

Finally, we can bring in the idea of flourishing. In our view, this usefully serves as the more all-encompassing notion, enfolding concepts such as well-being and health. To appreciate this system of conceptual nesting, we would argue that well-being and health are best seen as pertaining to singular living beings, with a central locus of sentience and agency, notably humans, but also animals and other conscious entities [58]. Even though a human might be understood as a symbiotic collective of micro-organisms, it fundamentally exists, operates, and is perceived as, a unified agent. By contrast, even though a city, say, is similarly a collective of organisms (people), it does *not* have a central locus of sentience or agency. Humans are part of numerous systems, both living (e.g., people constituting their community) and non-living (e.g., the economy). Moreover, as shown by a vast research literature, people’s health and well-being are greatly influenced by such systems or collectives, and the myriad factors that operate though these; such factors range from economic and political to cultural and ecological influences [59]. We view this whole dynamic scene—health and well-being, *and* the factors which influence these (many of which are systemic)—as together constituting flourishing. Thus, our definition of flourishing expands upon that of health and well-being above, constituting a state of personal and systemic quality in relation to all dimensions of existence, in a way that is relatively enduring and well-supported by the various conditions of life. An alternative more concise phrasing which captures the same idea is the relative attainment of a state in which all aspects of a person’s life are good, including the contexts in which that person lives [60].

Such definitions acknowledge that implicit in the concept of flourishing is the idea that the conditions of a person’s life are conducive to that flourishing, which is not necessarily the case with health and well-being. With those, it is possible and even not uncommon for people to attain these *in spite of* their context and circumstances. Similarly, the verb “to thrive” in some sense suggests well-being potentially regardless or even despite inhospitable conditions. This is reflected in its etymology, entering English around 1200 CE from a Scandinavian source akin to Old Norse *þrifask* (originally “grasp to oneself”). By contrast, flourishing implies being supported by one’s environment; etymologically, it directly relates to our central garden metaphor, and derives ultimately from the Latin *florere* (“to bloom, blossom, flower”). Thus, flourishing suggests adaptive interaction and consonance between the individual and their contextual systems, such that these help people within to prosper, and perhaps vice versa. To the latter point, flourishing not only implies that one’s conditions are conducive to well-being; it can also pertain to the systems and conditions themselves. Thus, one might speak of an economy or a community, say, as flourishing. Indeed, we are often also interested in these systems for their own sake (e.g., how well a community is faring).

We can again turn to our two metaphors to help bring these dynamics to life. With the garden metaphor, so far, we have focused on the individual human as a tree (or plant, flower, etc.). However, the broader context is that the person’s sociocultural environment is the garden in which the tree is situated, along with all their fellow humans, who similarly constitute flora in this vast ecology of humanity. In our view, a person might attain some measure of well-being, and even thrive, in spite of their environment. However, to truly flourish, so must their garden. Indeed, the reverse is also true: while the tree would struggle without the nurturing environment of the garden, the garden would be nothing without its flora. Flourishing means the person *and* their context—the tree *and* the garden—both doing well. Thus, human well-being is intrinsically connected to the environment in which people are situated. We use the term “environment” in its broadest possible sense. To begin with, humans are nested inextricably in networks of other humans: from family and friends to the broader community at various levels of scale [61,62]. A wealth of work shows that such networks and relationships may be the most important determinant of well-being [63]. Indeed, except for extreme examples (e.g., monastic hermits), humans literally cannot survive without the support and communion of their fellow humans. A lone tree or plant is an anomaly; they usually need networks of flora to survive. However, it is not just about other humans. In a very real and non-metaphorical sense, humans *do* exist in a vast garden of our global ecosystem [64]. Crucially, we cannot fully flourish if this ecosystem is struggling. First, the burgeoning literature shows that mental well-being is strongly influenced by environmental factors. These range from, in a positive sense, the value of proximity and access to green and blue spaces [65] to, in a negative sense, the detrimental impact of issues such as poor air quality [66], as well as the burden imposed by people’s concern about the environment, as reflected in emergent phenomena such as ecoanxiety [67]. However, the impending climate crisis is showing that, if environmental conditions worsen, not only may flourishing be imperilled, but also human existence itself—or, at least, civilized existence, involving a viable and functioning society. In a very real and urgent sense, we need our garden to flourish if we are to live at all.

Such considerations also bring up weighty and difficult issues. Among the critiques of PP is the charge of individualism—that it focuses primarily on a person’s individual well-being irrespective of its impact or consequences for others [68]. However, our conception of flourishing demands one wrestles with how a person is doing in the broader context of other people and their sociocultural environment. In our garden metaphor, what happens if a particular plant (i.e., person) is a poisonous weed (e.g., a violent criminal) who not only encroaches upon but endangers plants around him? Moreover, what if the growth of this noxious plant constitutes *its* own version of thriving? We would not call it flourishing, since a path of maximal self-interest that is detrimental to other people and one’s broader context does not constitute flourishing in our view, which involves both a person and their environment doing well. However, that weed could potentially still be said to be pursuing its own idiosyncratic and solipsistic well-being in some way. A criminal who takes pleasure in violence could be seen as at least experiencing some forms of well-being—e.g., hedonic enjoyment—even if at the expense of other forms. In such cases, what rights or responsibilities do those with power (as “gardener”) have to restrain that plant for the good of other plants and the entire garden, and even for the plant’s *own* overall flourishing? Similarly, what rights and freedoms do people have to pursue forms of well-being which they value but harm those around? These are weighty and thorny issues that have perplexed philosophical and legal minds for centuries. It is beyond our scope to delve into let alone resolve these here, but this metaphor—and this broader conception of flourishing—does at least bring these to attention. 

Similar considerations are raised by our orchestral metaphor. In terms of well-being, we have focused on the sound of a given instrument. However, this instrument is but one sound among many in the overall symphony of humanity. The instrument must peacefully and productively co-exist alongside the sounds of all other instruments—i.e., fellow humans—in the orchestra. This does not mean losing its identity or freedom in a form of totalitarian uniformity or pre-determined arrangement. Although real-life orchestras do tend to involve whole sections of instruments playing in lock-step from a fixed composition, in our metaphor, there is much more scope for autonomy and agency, innovation and improvisation (e.g., one might think of a jazz orchestra in particular). However, the sounds still need to gel harmoniously together. Otherwise, there is just chaos and cacophony, a discordant strife. Similarly, in human affairs, people have considerable latitude in their choices and actions, but this does not mean an unbridled free-for-all; we need to abide by common norms and laws for civilization to be possible. Then, besides these considerations of harmonising with our fellow humans, the orchestra metaphor also recognises the importance of all the myriad societal factors that influence well-being. After all, the orchestra’s success depends not only on the sounds it produces, but on all the supporting conditions that make it possible and viable in the first place. These range from how well the orchestra is funded and managed, to the support and behaviour of its patrons and audience, to the kind of society that even allows and encourages live music in the first place. An individual instrument may play a beautiful tune, but it needs all these factors in place to truly flourish, and indeed perhaps to even exist at all. Thus, flourishing refers not only to the quality of the instrument’s sound, but also to the entire orchestra—and its supporting conditions—in which it is given a voice. And so too with our human flourishing.

## 6. Conclusions

This paper has explored the way in which our understanding of well-being and related ideas, such as health and flourishing, are shaped by the metaphors through which we think about them. We began by suggesting that the current dominant metaphors—including a pyramid, ladder, and continuum—are all limited and problematic in various ways. In their unidimensionality, for example, they struggle to account for the nuances of well-being, while in their individuality, they also do not allow sufficient appreciation for the way in which well-being is influenced by contextual factors. As such, we offered two other metaphors which we suggest do better justice to the dynamic complexities of these concepts, namely, a garden and an orchestra. Through these metaphors, we sought to articulate a comprehensive framework for conceptualizing and appreciating the nature of well-being and related concepts. We hope these metaphors will prove fruitful and generative in inspiring further insights and research into these most-valued and sought-after phenomena.

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
