# Peer review of "The Garden and the Orchestra: Generative Metaphors for Conceptualizing the Complexities of Well-Being"

_ijerph, 2022, doi:10.3390/ijerph192114544_

Round 1
Reviewer 1 Report
This paper analyzes the nature of happiness. First, the metaphors such asa pyramid, ladder, and continuum based on the previous studies are carefully analyzed, and the existing problems and limitations of these metaphors are analyzed. Then, this paper provides two other metaphors, namely a garden and an orchestra, to illustrate the subtle complexity of such concepts of well-being, health, and flourishing. Through these metaphors, this paper articulates a comprehensive framework for conceptualizing and appreciating the nature of well-being (and associated concepts). This paper is very detailed and clear, and it very well helps people to understand the nature of such concepts as well-being, health, and flourishing.
As far as the quality of the article is concerned, it is acceptable. However, I am not sure whether this kind of article is the scope of the journal, and I am not sure whether the readers will be interested in this kind of article, which needs to be confirmed by the editor.
Author Response
Thank you for your positive comments.
Reviewer 2 Report
i am grateful for the opportunity to review your manuscript.
i offer the following for your consideration:
- do consider a little elaboration of the heading for the opening section of your manuscript. currently, it reads "a metaphorical approach". having read the entire manuscript through, i believe i understand your economy of words. however, it is your opportunity to make a strong first impression and to lead your reader in to the remainder of your manuscript, rather than to leave it a little vague. i leave it to you to consider the merits of either. i believe your original three words can remain, but do consider just a little elaboration (eg, to what?)
- you end the second paragraph in this section with "Given that, ...". do consider expressing this as "Given the above", or, "Given the preceding arguments" (or similar).
- when you first introduce the spectrum metaphor, you have a clause "its adherents ..." the use of the possessive pronoun here is a little vague, as there were at least two possible subjects immediately preceding (the metaphor itself, or, PP).
- in the second paragraph of the section 'new well-being metaphors', the sentence "Thus, one could imagine..." is a little awkwardly and vaguely expressed. do you mean to say, for instance, that "one could imagine all these different aspects .. as each being a separate branch of the tree"? (in which case, please add 'each being' (or similar). or, do you possibly mean that all these different aspects are indeed just a single separate branch? as you take pains to point out, the metaphor can be as granular as one wishes, but i believe you might have a specific degree of granularity in mind, at this juncture of your argument.
Author Response
Thank you for your positive comments and constructive feedback. Our response to your comments can be found below in italics.
Do consider a little elaboration of the heading for the opening section of your manuscript. currently, it reads "a metaphorical approach". having read the entire manuscript through, i believe i understand your economy of words. however, it is your opportunity to make a strong first impression and to lead your reader in to the remainder of your manuscript, rather than to leave it a little vague. i leave it to you to consider the merits of either. i believe your original three words can remain, but do consider just a little elaboration (eg, to what?)
Thank you for the suggestion. We have amended this to "A Metaphorical Approach to Conceptualizing Wellbeing".
You end the second paragraph in this section with "Given that, ...". do consider expressing this as "Given the above", or, "Given the preceding arguments" (or similar).
We have now rephrased this to "Given these insights".
When you first introduce the spectrum metaphor, you have a clause "its adherents ..." the use of the possessive pronoun here is a little vague, as there were at least two possible subjects immediately preceding (the metaphor itself, or, PP).
We have now rephrased this to "the adherents of this new field".
In the second paragraph of the section 'new well-being metaphors', the sentence "Thus, one could imagine..." is a little awkwardly and vaguely expressed. do you mean to say, for instance, that "one could imagine all these different aspects .. as each being a separate branch of the tree"? (in which case, please add 'each being' (or similar). or, do you possibly mean that all these different aspects are indeed just a single separate branch? as you take pains to point out, the metaphor can be as granular as one wishes, but i believe you might have a specific degree of granularity in mind, at this juncture of your argument.
We have now rephrased this sentence to "All these different aspects of mental ill-being and well-being could be imagined as each constituting a separate branch of the tree.".
Reviewer 3 Report
I believe it is a good proposal although it may be complicated for people not related to music (first metaphor). It may help a bit of context, perhaps an illustration.
Author Response
I believe it is a good proposal although it may be complicated for people not related to music (first metaphor). It may help a bit of context, perhaps an illustration.
Thank you for your positive comment. Regarding the metaphor, although we believe most readers will easily grasp the concept, for those unfamiliar with the nature of an orchestra we have added a sentence when we first introduce it (on p.7) clarifying what it involves.